# Contributions of human amygdala nuclei to resting-state networks

Uriel K. A. Elvira[1,5,6], Sara Seoane[1,5,6], Joost Janssen[2,3,4], Niels Janssen[1,5,6,7]*

**1** Department of Psychology, Universidad de la Laguna, Santa Cruz de Tenerife, Spain, **2** Department of Child and Adolescent Psychiatry, Institute of Psychiatry and Mental Health, Hospital General Universitario Gregorio Marañón, Madrid, Spain, **3** Ciber del Área de Salud Mental, Instituto de Investigación Sanitaria Gregorio Marañón, Madrid, Spain, **4** Department of Psychiatry, UMCU Brain Center, University Medical Center Utrecht, Utrecht, The Netherlands, **5** Institute of Biomedical Technologies, Universidad de La Laguna, Santa Cruz de Tenerife, Spain, **6** Institute of Neurosciences, Universidad de la Laguna, Santa Cruz de Tenerife, Spain, **7** Department of Neurobiology and Behavior, University of California, Irvine, California, United States of America

* njanssen@ull.es

**Data Availability Statement:** All data are freely available from the Human Connectome Project database (http://db.humanconnectome.org/). Further supporting data is available at: https://github.com/iamnielsjanssen/AmygdalaFC.

## Abstract

The amygdala is a brain region with a complex internal structure that is associated with psychiatric disease. Methodological limitations have complicated the study of the internal structure of the amygdala in humans. In the current study we examined the functional connectivity between nine amygdaloid nuclei and existing resting-state networks using a high spatial-resolution fMRI dataset. Using data-driven analysis techniques we found that there were three main clusters inside the amygdala that correlated with the somatomotor, ventral attention and default mode networks. In addition, we found that each resting-state networks depended on a specific configuration of amygdaloid nuclei. Finally, we found that co-activity in the cortical-nucleus increased with the severity of self-rated fear in participants. These results highlight the complex nature of amygdaloid connectivity that is not confined to traditional large-scale divisions, implicates specific configurations of nuclei with certain resting-state networks and highlights the potential clinical relevance of the cortical-nucleus in future studies of the human amygdala.

## Introduction

The amygdala is a brain structure located in anterior sections of the medial temporal lobe. Interest in this region stems from its association with a number of common psychiatric disorders such as depression and schizophrenia [1–6]. The amygdala consists of a number of interconnected nuclei, each of which is thought to contribute to different aspects of emotion and cognition [7–9]. For example, animal studies have suggested that the lateral nucleus of the amygdala (La) plays a key role in fear learning, while the central nucleus (Ce) is involved in fear responses [10–12]. However, methodological limitations in the spatial resolution of current neuroimaging techniques have hampered the study of the amygdaloid nuclei in humans [13, 14]. Testing the generalizability of animal findings would improve our understanding of amygdala function and aid in the translation to clinical applications [15, 16]. Here we focused

**Funding:** This work was supported by the Human Connectome Project, WU-Minn Consortium (1U54MH091657) funded by the 16 NIH Institutes and Centers that support the NIH Blueprint for Neuroscience Research; and by the McDonnell Center for Systems Neuroscience at Washington University. This work was also supported by the Ministerio de Ciencia e Innovación via grants PSI2017-84933-P and PSI2017-91955-EXP awarded to NJ. This work was also supported by the Board of Economy, Industry, Trade and Knowledge of the Canarian Government with a European Social Fund co-financing rate via grant TESIS2019010146 awarded to SS. The funders had no role in study design, data collection and analysis, decision to publish, or preparation of the manuscript.

**Competing interests:** The authors have declared that no competing interests exist.

on the functional connectivity (FC) of the amygdaloid nuclei using the very high spatial-resolution 7T resting-state functional Magnetic Resonance Imaging (rsfMRI) dataset from the Human Connectome Project (HCP). Our focus was on discovering how the amygdaloid nuclei were connected to known resting-state networks. We also examined how the FC of the individual nuclei was associated with participants' ratings of self-reported fear.

There are thirteen different amygdaloid nuclei (see Table 1 and Fig 1 for an overview of the nuclei examined here). Previous tract-tracing studies in non-human animals have elucidated the anatomical connectivity of the amygdala [17]. The classical view is that primary sensory areas project mainly to the La nucleus, whereas the Ce nucleus projects to striatal and motor output regions [10]. In addition, it is known that other brain regions have more complex anatomical connections with the nuclei of the amygdala. For example, the prefrontal cortex, an area thought to play an important role in modulating activity in the amygdala, is known to project to basal (Ba), La, accessory basal (AB), Ce and medial (Me) nuclei [18–20]. Similarly, both the entorhinal cortex and hippocampus are known to project to all amygdala nuclei [21, 22]. In humans, investigation of the FC of the amygdala has relied on low spatial-resolution fMRI acquisition protocols (around 2–4 mm) which do not permit the study of the individual amygdaloid nuclei. Instead, human FC studies have attempted to establish the connectivity between the amygdala and the rest of the brain based on large-scale sections of the amygdala that aggregate across individual nuclei. The most frequently used amygdala parcellation is based on its cytoarchitectonic properties and proposes that there are three large-scale sections called LateroBasal (LB), CentroMedial (CM) and SuperFicial (SF; [23], see Table 1 for an overview).

Previous studies have attempted to link the whole amygdala as well as its three large-scale sections with whole-brain resting-state networks such as those of [24]. However, these studies have not produced clear results. Specifically, whereas studies have found FC between the amygdala and regions of the default mode, somatomotor, and (dorsal/ventral) attention networks [6, 25–27], these networks were not consistently detected across all studies. Thus, for example, while regions of the default mode network co-activated with the amygdala in some studies [6, 25, 26], other studies did not find amygdala co-activity with the regions of the default mode network [27]. In addition, how the large-scale divisions of the amygdala co-activated with these networks remains unclear. For example, whereas some studies have found that the LB region was associated with the default mode network [25], others have concluded that the SF region co-activated with this network [26], and yet others have suggested the CM region is linked with the default mode network [6]. Thus, overall, these previous studies have not examined FC of the amygdala nuclei due to their reliance on large-scale parcellations of the amygdala.

**Table 1. Amygdala subnulcei and their abbreviations.**

| Whole name | Abbreviation | Large-scale division |
|---|---|---|
| Accessory Basal nucleus | AB | LB |
| Basal nucleus | Ba | LB |
| Lateral nucleus | La | LB |
| Paralaminar nucleus | PL | LB |
| Central nucleus | Ce | CM |
| Medial nucleus | Me | CM |
| Anterior Amygdaloid Area | AAA | SF |
| Cortical nucleus | Co | SF |
| CorticoAmygdaloid Transition area | CAT | SF |

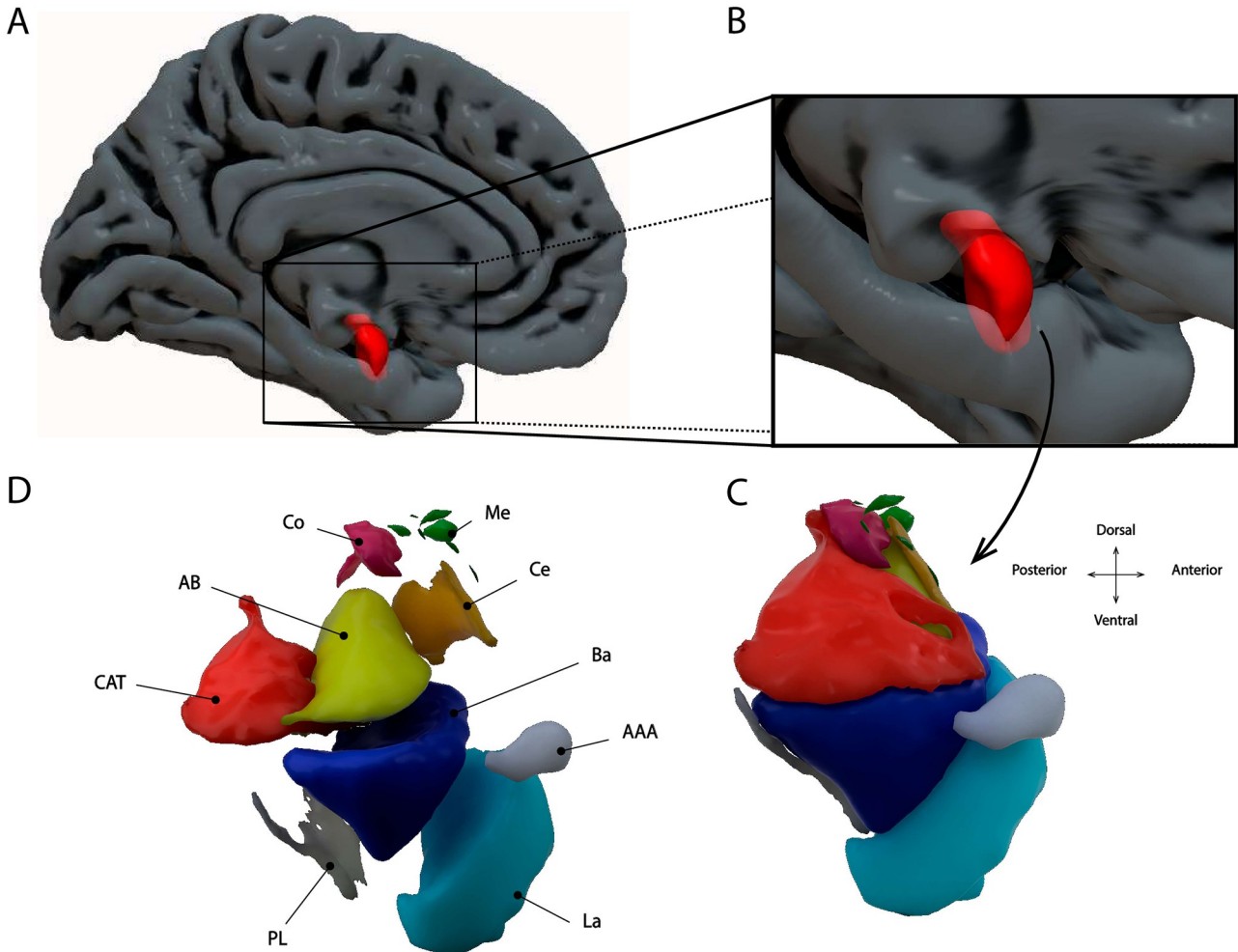

**Fig 1. Overview of the location of the amygdala.** A,B: Amygdala in a medial view of the brain. C,D: The location of the nine different amygdaloid nuclei within the amygdala.

Here we examined the FC of the different amygdaloid nuclei with the rest of the brain using the resting-state fMRI (rsfMRI) dataset from the Human Connectome Project (HCP) with 1.6 mm isotropic resolution. We used a data-driven analysis approach to examine how different amygdaloid nuclei contributed to various reference resting-state networks [24]. In addition, we tested whether the self-reported fear of the participants was associated with co-activity in the amygdaloid nuclei. Finally, we relied on a test-validation approach in which results obtained in the test dataset were validated in a separate dataset.

## Materials and methods

### Participants

All data were publicly available from the HCP website. Within this large dataset with over 1100 participants, there were 184 participants that had rsfMRI data with 1.6 mm spatial resolution that was collected at 7T. From this initial set we excluded participants that did not meet HCP specified Quality Control (QC) issues. Specifically, we removed 12 participants that did not meet QC issue A (anatomical anomalies), B (segmentation/surface errors), C (head coil

instabilities), and D (significant resting-state artefacts) leading to a total dataset of 172 participants. Within this final dataset, 68 participants were male, and the most common age-range was 26–30 years (81 participants). The study was approved by the Research Ethics and Animal Welfare Committee (CEIBA) at the University of La Laguna (CEIBA2017–0270). Data were fully anonymized before they were accessed and therefore no consent was obtained.

## MRI data acquisition

As per the HCP reference manual, the 7T data was collected on a Siemens Magnetom scanner located at the Center for Magnetic Resonance (CMRR) at University of Minnesota in Minneapolis, MN. The scanner uses the Nova32 32-channel Siemens receive head coil with an incorporated head-only transmit coil that surrounds the receive head coil from Nova Medical. Volumes were acquired using Gradient-Echo EPI. Each volume contained 85 slices that were acquired with a multiband factor of 5. Slice thickness was 1.6 mm with no gap, the FOV was 208 x 208 mm, matrix size 130 x 130, resulting in 1.6 mm isotropic voxels. The in-plane acceleration parameter (iPAT) was 2. The TR was 1000 ms, echo time (TE) 22.2 ms, and the flip angle 45˚. In each run 900 volumes were collected and lasted around 16 minutes. Runs alternated between phase encoding in the posterior-anterior (PA) and anterior-posterior (AP) direction. Matched phase reversed spin-echo fieldmaps were collected and used for distortion correction. There were four rsfMRI sessions available for each participant. For our test-validation design, we used data from the first two sessions with alternating phase encoding directions as a test dataset, and the remaining two sessions as a validation dataset.

In addition, structural T1w and T2w images were available for each participant. Again as per the HCP reference manual, these images were acquired on a customized Siemens Connectom Skyra 3T scanner. The T1w images were acquired using a 3DMPRAGE protocol TI/TR/TE: 1000/2400/2.14 ms, flip angle = 8˚, resulting in 0.7 mm isotropic voxels. The T2w images were acquired using a 3D T2-SPACE protocol TR/TE: 3200/565 ms, flip angle = variable, and also resulting in 0.7 mm isotropic voxels.

## Preprocessing

We downloaded the rsfMRI 1.6mm/32k FIX-Denoised (Compact) and rsfMRI FIX-Denoised (Extended) datasets for each participant. These datasets consist of already pre-processed functional data according to HCP minimal preprocessing pipelines ([28], see also Fig 2). Briefly, transformations that reduce head motion are estimated using FSL MCFlirt, fieldmap (matched phase reversed spin-echo) and gradient distortion corrections are applied, and transformations from fMRI space to MNI space are estimated using non-linear transformations. Importantly, any imposed spatial smoothing of the data was minimized in two ways: First, the transformation from native to MNI space preserved the native space resolution of the fMRI acquisitions, and second, all transformations were postponed, combined and applied in a single step using spline interpolation (see [28] for further discussion of this issue).

Next, the data in MNI space was temporally filtered using a 2000 s highpass filter and automatically denoised using the FIX program [29, 30]. This program uses semi-automatic classification of head-motion and other artifacts which minimized the potential impact of head-motion artefacts in our data. The final files were demeaned and had native 1.6mm isotropic resolution in MNI space. In addition, we extracted cerebrospinal fluid (CSF) and white-matter masks from each participant's freesurfer parcelation (in the wmparc file), obtained their timeseries by averaging across all voxels inside each mask, and then regressed this timeseries out of each participant's fMRI data. The residual timeseries were then used as the final cleaned data on which further downstream analyses were performed.

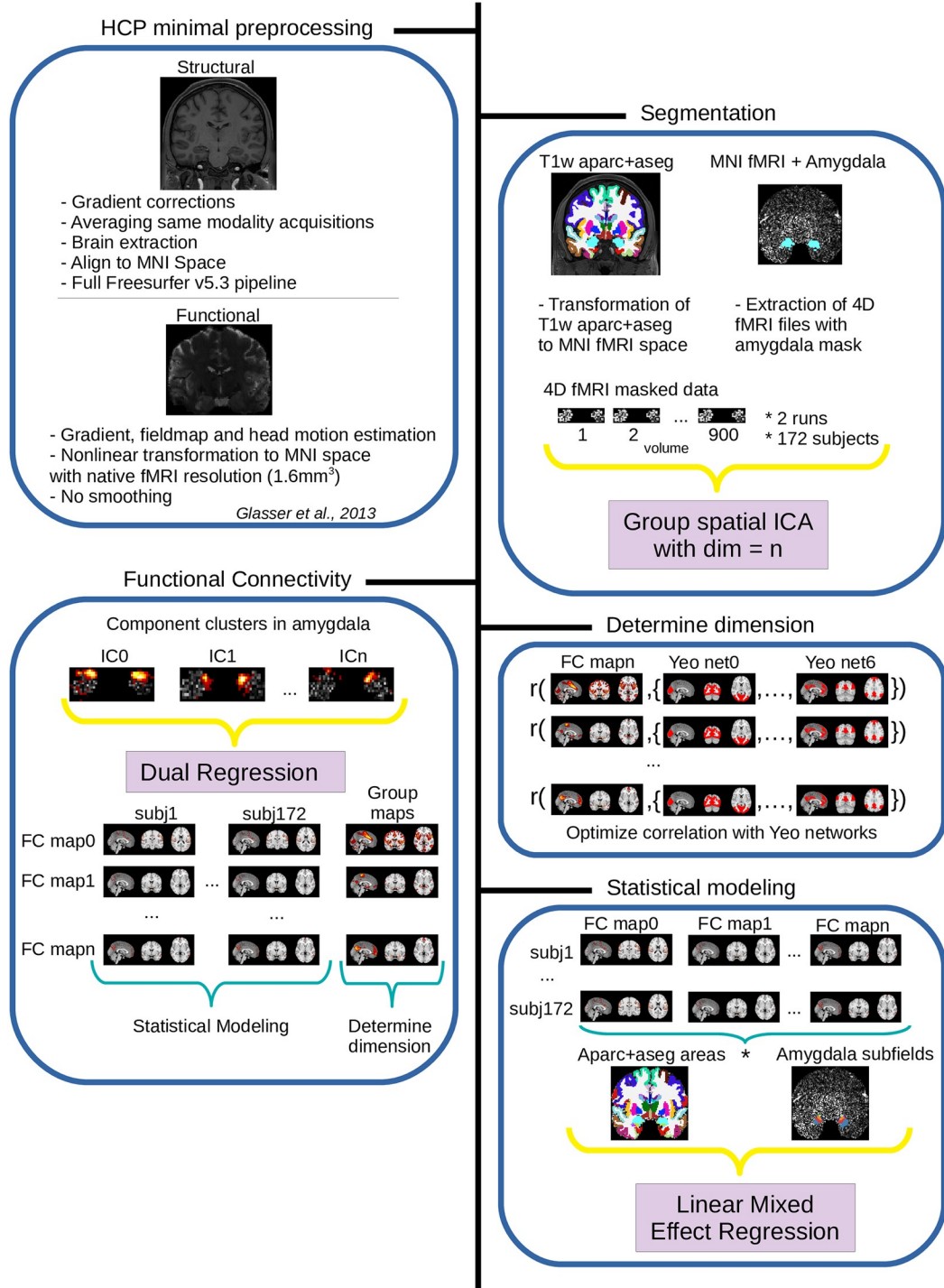

**Fig 2. Overview of the analysis pipeline in our data.** See text for details.

For the structural data, we downloaded the 3T Structural Preprocessed and 3T Structural Preprocessed Extended packages. These packages contained the T1w and T2w images for each participant as well as the full Freesurfer output and transformation matrices that were relevant for our downstream analyses (see below). For specific information on the pre-processing of these structural images we refer to [28].

## Analyses

The main aim of the study was to explore the relative contributions of the amygdala nuclei to various resting-state networks. To obtain this objective the analyses were divided into six main parts.

**Segmentation of amygdaloid nuclei.** First, for each participant we had available the full output produced by Freesurfer v5.3 [31]. Automatic segmentation of the amygdaloid nuclei was performed on this output by using the Hippocampal Subfields and Nuclei of the Amygdala script (v21) with the 0.7mm T2w image as the input [9]. This script produces an automatic segmentation of the nine different nuclei of the amygdala (see Fig 1 for a visual presentation). The nine different nuclei were the anterior amygdaloid area (AAA), cortico-amygdaloid transition area (CAT), basal nucleus (Ba), lateral nucleus (La), accessory basal nucleus (AB), central nucleus (Ce), cortical nucleus (Co), medial nucleus (Me), and paralaminar nuclus (PL; See Table 1 for an overview and Fig 1 for a graphical presentation of the nuclei). These subdivisions do not take into account the concept of the extended amygdala, meaning that it excludes a basal forebrain structure called the bed nucleus of the stria terminalis [32]. The method proposed by [9] finds a segmentation for the amygdaloid nuclei in a given participant by combining prior information about the average structural location of the amygdaloid nuclei and its surrounding tissue with specific structural information (i.e., the relative position of structures and T1w signal intensity values) from a given participant's brain [33]. This method fundamentally improves over the more common method in which the segmentation of the amygdala takes place in a standardized (MNI) space (e.g., [23]). This method therefore allows for a probabilistic segmentation of nine amygdala nuclei that is sensitive to the unique morphology of an individual's brain.

**Detection of signal clusters in the amygdala.** The second step of the analyses relied on the detection of clusters inside the amygdala using spatially-restricted Independent Component Analysis (srICA). srICA is an emerging technique in which the ICA technique is applied to a restricted region of the central nervous system [34–37]. Previous studies have shown that srICA leads to a more sensitive detection of signals inside the restricted brain region compared to whole-brain ICA [38, 39]. In the current study we applied group ICA using melodic v3.15 [40] with default options (except MIGP off) to the concatenation of 344 fMRI datasets (172 participants * 2 resting-state datasets) that were masked with a bilateral amygdala mask. As in other studies from our group [41, 42], classification of detected Independent Components (ICs; referred to here as 'seed clusters' or ICs) inside the amygdala as signal or noise proceeded in three steps. First, group level whole-brain FC maps that were correlated with each detected seed cluster were computed using Dual Regression and FSL randomise [43]. Second, each group level whole-brain FC map linked to a specific amygdala seed was correlated with each of the 7 reference resting-state networks of [24]. Third, this procedure was optimized across a wide range of ICA dimensions such that it found the dimension at which the correlations between the obtained whole-brain FC maps and the reference networks was highest. This procedure therefore detected in a data-driven fashion the dimension for which the srICA produced voxel clusters inside the amygdala that were maximally associated with known resting-state networks (please see Fig 2 or [41, 42] for further information).

**Group-level analyses of whole-brain FC.** The next step served to obtain lists of brain regions involved in each whole-brain resting-state network. We relied on a parcel level approach in which we extracted the average Z-value for each brain region listed in the Desikan-Killany atlas from each participant-specific whole-brain FC map. These data were then subjected to a regression model of the form:

$$Z = hemisphere + FC\_map \times brain\_region + rand(participant), \qquad (1)$$

where *hemisphere* was a discrete co-variable with two levels (left vs right), *FC_map* was a factor with number of levels equal to the number of ICs with neural origin (i.e., here 3), *brain_region* was a factor with number of levels equal to the sum of the number of cortical and subcortical regions in the aparc+aseg atlas, and *participant* was a random effect with number of levels equal to the total number of participants (i.e., 172). The dependent variable *Z* was the average Z-value for the cortical and subcortical regions obtained from each participant's aparc+aseg file. Note that in this model we computed a random-effect for participant that takes into account the highly probable between-participant variability in the FC maps.

Within this model, our main interest was in the interaction term *FC_map × brain_region*. This interaction provided a test of the null-hypothesis that the different FC maps would yield the same average Z values across the cortical and subcortical brain regions. In the case that the interaction term was significant, we performed post-hoc comparisons where we compared the FC map associated with a given IC to the mean of the other FC maps (i.e., an "effect" contrast). This therefore produced a list of cortical and subcortical regions that were more co-activated in one FC map versus the other FC maps.

**Relative contributions of amygdaloid nuclei.** The next step of the analysis was to determine the relative contribution of the individual amygdaloid subnuclei in the various whole-brain resting-state networks identified in the previous step. Participant-specific amygdaloid subnuclei masks were intersected with each participant-specific whole-brain FC map. The resulting average Z-values for each amygdala nucleus and each participant were then fitted to the same statistical model as described in Eq 1. In this model, the term *brain_region* now referred to the nine amygdaloid nuclei. As before, our specific interest was in the interaction term of the model (*FC × brain_region*). To determine the relative contribution of the subnuclei to the different resting-state networks, pairwise comparisons of all nine amygdaloid nuclei within each FC map were performed. The pairwise test-statistics were then summed, ordered, and thresholded at > 0. This therefore produced a ranked-estimate of the relative contribution of each amygdaloid nucleus to each whole-brain FC map.

**Validation analysis.** To validate the results, we examined the degree to which the whole-brain FC maps associated with the amygdala clusters we obtained in the test dataset generalized to the validation dataset. We quantified this step by computing the correlation between the whole brain FC maps in the test and validation sets, and by comparing the correlations of the whole brain FC maps with the reference networks of [24] in the test and validation sets.

**Impact of self-reported fear on amygdaloid nucleus co-activity.** Finally, we explored whether self-reported measures of fear associated with co-activity of the amygdala nuclei. Self-reported measures of fear were obtained from the Fear Affect CAT 18+ questionnaire that is part of the NIH Toolbox [44]. This questionnaire contains a set of 29 items where participants rated questions relating to experienced fear/anxiety over the past 7 days on a scale from 1 (never) to 5 (always; e.g., "I felt frightened. . .1–2-3-4–5"). Previous studies have provided further assessment and validation of this test [45]. The 172 participants in our sample were then classified into groups of low and high fearful participants on the basis of a median split of their Fear Affect scores (see Table 2). These groups subsequently differed in terms of their fear affect scores (t(170) = 15.47, p < 0.0001).

**Table 2. Description of the sample.**

| Group | N | Mean score (SD) | N women |
|---|---|---|---|
| low-fear | 86 | 44.0 (4.9) | 47 |
| high-fear | 86 | 55.2 (4.5) | 57 |

To examine whether amygdala subnucleus FC associated with by self reported fear we modified the model in Eq 1 to include variables for *Age* and *Gender* as well as a three-way interaction for *Group × FC_map × brain_region*. Our main interest in this model was in the interaction term of *Group × FC_map × brain_region* and *Group × brain_region*. If this interaction was significant, we pairwise compared the low fear group versus the high fear group for each nucleus and each FC map at the same time controlling for multiple corrections using bonferroni correction. This analysis therefore produced a list of those amygdala nuclei whose co-activity values were sensitive to changes in self-reported fear.

All statistical modeling took place in the statistical computing environment R (v4.0.0). Mixed effect modeling relied on the lme4 package (v1.1.23) [46]. Results from these regression models are presented in the form of type III ANOVA tables that were computed directly from the output of the mixed effect models using the lmerTest package (v3.1–2) [47]. P-values in these models were computed using the Saitherwaite correction for the degrees of freedom. Posthoc testing was performed using the emmeans package (v1.4.6) [48] when a given interaction term was significant (i.e., $p < 0.05$). P-values in these posthoc tests were adjusted for multiple comparisons using the Bonferroni method. We visualized these results using the ggseg (v1.5.4) [49], and ggpubr (v0.3.0) packages [50].

## Results

### Detection of signal clusters in the amygdala

The procedure for finding the optimal number of dimensions first returned that across the tested ICA dimensions 1 to 15, obtained group level FC maps correlated strongly ($r > 0.4$) with the somatomotor, ventral attention, and default mode networks. Furthermore, our optimization method found that the smallest dimension at which these 3 networks were strongly and uniquely detected was for ICA with dimension = 7. Specifically, we found that for this specific ICA dimension, IC2 correlated strongly and uniquely with the ventral attention network ($r = 0.48$), IC3 with somatomotor network ($r = .45$), and IC5 with the default mode network ($r = 0.59$; see Table 3 for an overview of the correlations for each IC with all 7 networks). In addition, as can be seen in S1 Fig, strong correlations ($r > 0.4$) were frequently found for these three networks in other dimensions, suggesting that the detection of these three networks was not idiosyncratic to dimension 7. Similarly, as can be seen in S2 Fig, ICA at dimensions 20 and 30 did not lead to the detection of new networks. We can therefore conclude that the specific clusters of voxels detected by the ICA using dimension 7 for IC2, IC3, and IC5 were optimal in connecting with three known resting-state networks.

A visual presentation of the location of the three seed clusters detected inside the amygdala along with their associated whole-brain group-level FC map is presented in Fig 3. A further overview of all 7 seed clusters with their corresponding FC maps is shown in S3 Fig. As can be

**Table 3. Table of correlations of the whole-brain FC maps for each IC with known resting-state networks.**

|  | IC2 | IC3 | IC5 |
|---|---|---|---|
| Visual | 0.16 | 0.25 | 0.01 |
| Somatomotor | 0.31 | 0.45 | 0.02 |
| Dorsal Attention | 0.23 | 0.10 | 0.01 |
| Ventral Attention | 0.48 | 0.01 | 0.01 |
| Limbic | 0.01 | 0.02 | 0.02 |
| Executive Control | 0.10 | 0.01 | 0.01 |
| Default Mode | 0.01 | 0.09 | 0.59 |

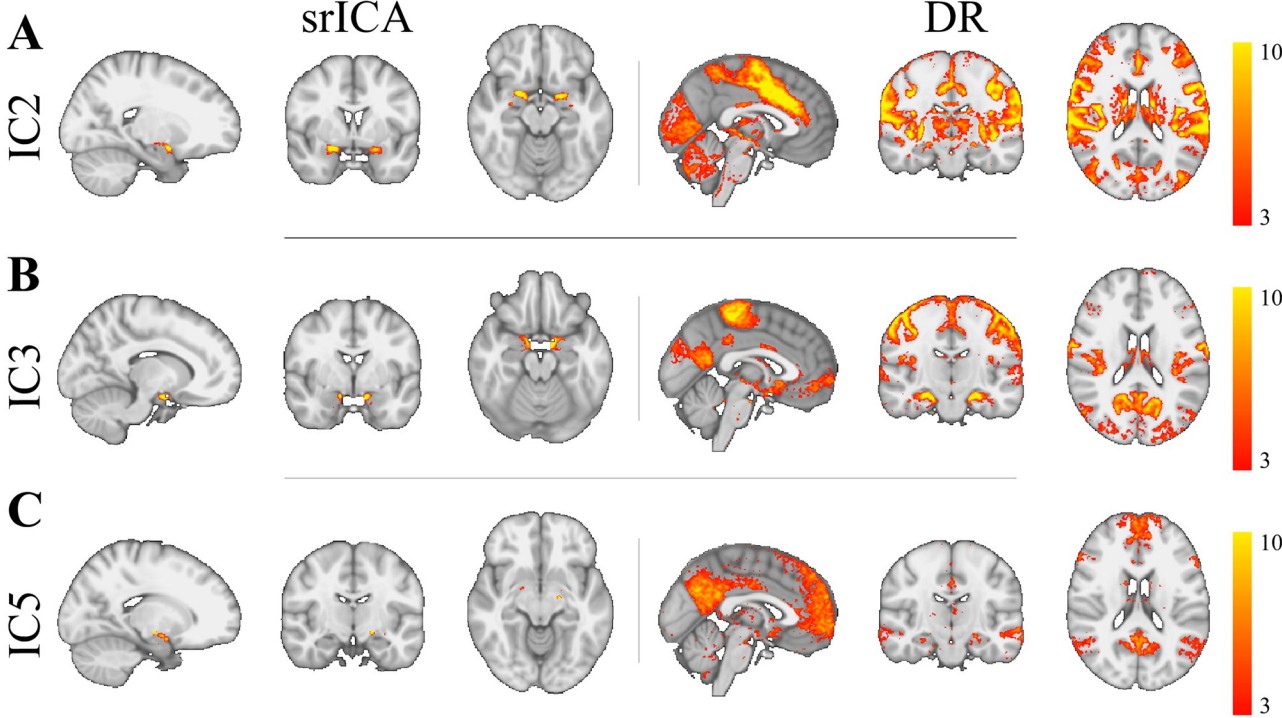

**Fig 3. Results from srICA and dual regression.** srICA left colum and Dual Regression right column. A: IC2 (Ventral Attention), B: IC3 (Somatomotor), C: IC5 (Default Mode). Note how different IC hotspots inside the amygdala connect to separate whole brain networks.

seen in Fig 3, the three seed clusters are located at different positions in the amygdala and revealed contrasting FC with the rest of the brain, indicating the involvement in different whole-brain networks. A more detailed view of the location of each seed cluster within the amygdala can be seen in Fig 4A–4C, where it can be seen that the three clusters detected by spatially restricted ICA occupy positions within the amygdala that likely reflect activity across multiple nuclei. How these activation clusters are connected with the rest of the brain as well as how they are distributed across the various amygdaloid nuclei will be examined in more detail below.

## Group-level analyses of whole-brain FC

Group-level analyses revealed the specific lists of regions co-activated in each FC map (see Fig 5 for the average Z values for each region by FC map). Specifically, multiple regression analyses revealed a main effect of Hemisphere ($F_{(1,44588)}=6.35$, $p < 0.02$), suggesting higher co-activity values in the left versus the right hemisphere. In addition, there was a main effect of Brain Region ($F_{(43,44588)}=64.45$, $p < 0.0001$) suggesting that co-activity values differed between the different cortical and subcortical brain regions. Furthermore, there was a main effect of FC map ($F_{(2,44588)}=1550.54$, $p < 0.0001$), suggesting co-activity values differed between the different FC maps. Important for our present purposes, there was a significant interaction between Brain Region and FC map ($F_{(86,44588)}=147.05$, $p < 0.0001$) suggesting that average co-activity values for each brain region differed between the different FC maps. Further exploring this interaction with posthoc tests revealed the list of regions where one FC map was significantly more co-activated compared to the other two maps. As can be seen in Table 4, FC map corresponding to IC2 revealed regions typically associated with the ventral attention

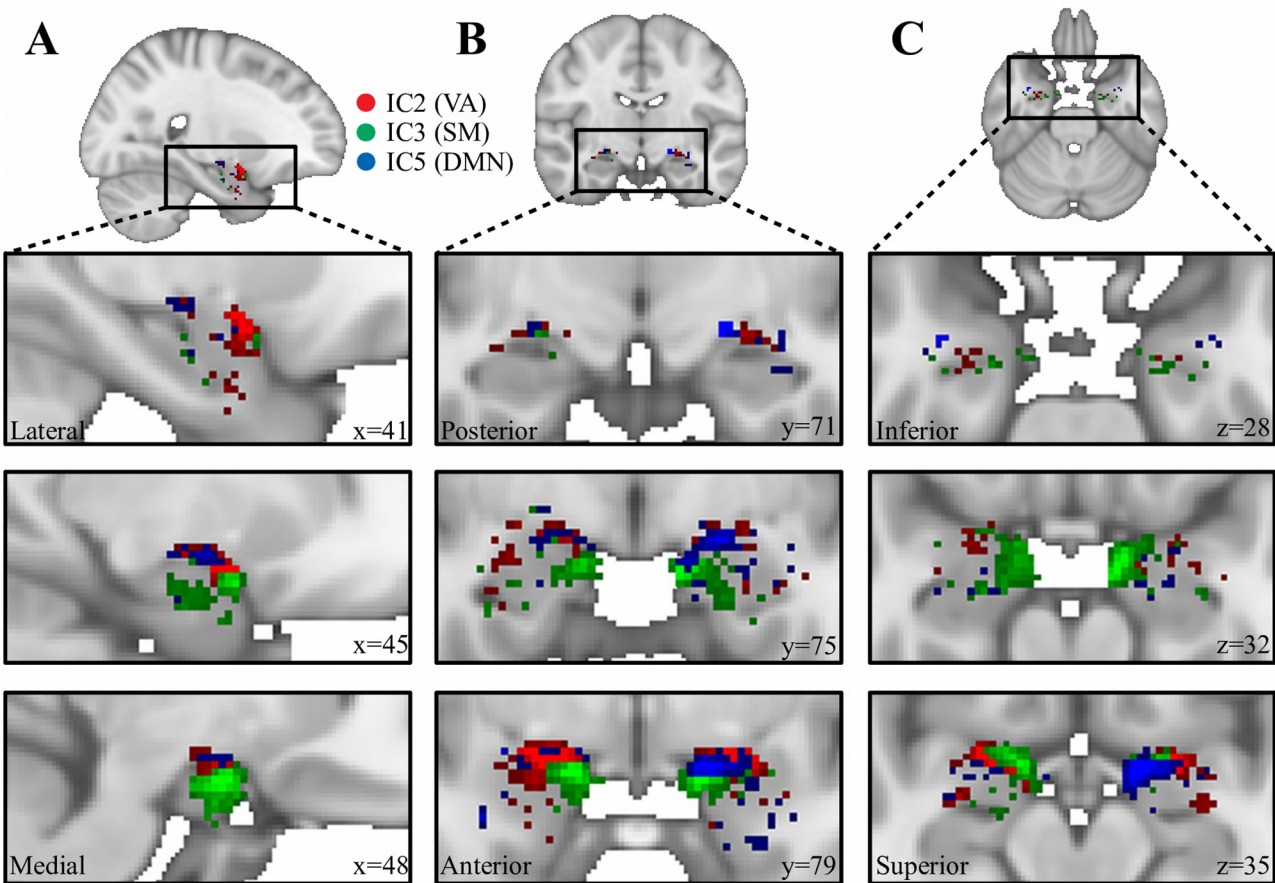

**Fig 4. Detailed view of results from srICA.** C2 (red, associated with Ventral Attention), IC3 (green, associated with Somatomotor) and IC5 (blue, associated with Default Mode) in saggital (A), coronal (B), and axial (C) views.

network like supramarginal gyrus, insular and anterior cingulate cortices. In addition, Table 5 showed that FC map corresponding to IC3 had high co-activity values in areas typically associated with the somatomotor network like sensorimotor cortex (post and paracentral gyrus). Finally, Table 6 revealed that FC map corresponding to IC5 had strong co-activity values in regions typically associated with the default mode network like the retrosplenial cortex (isthmus cingulate), precuneus, and medial orbitofrontal gyrus.

## Relative contributions of amygdaloid nuclei

Statistical analyses of the relative contribution of the amygdaloid nuclei in each of the FC maps revealed a main effect of Hemisphere ($F(1,9080.6) = 296.54$, $p < 0.0001$), suggesting increased co-activity values in the amygdala nuclei for the right than the left hemisphere. In addition, there was a main effect of Brain Region ($F(8,9089)=396.38$, $p < 0.0001$), suggesting that there were differences in co-activity between the nuclei. Furthermore, a main effect of FC maps was found ($F(2,9089)=1064.91$, $p < 0.0001$), suggesting differences in co-activity between the different whole brain FC maps. Again, important for our present purposes, there was a significant interaction between Brain Region and FC map ($F(16,9089)=395.55$, $p < 0.0001$), indicating that the nuclei were not co-activated in the same way across the different FC maps. Further exploration of this interaction using pairwise testing within each FC map and then ranking the

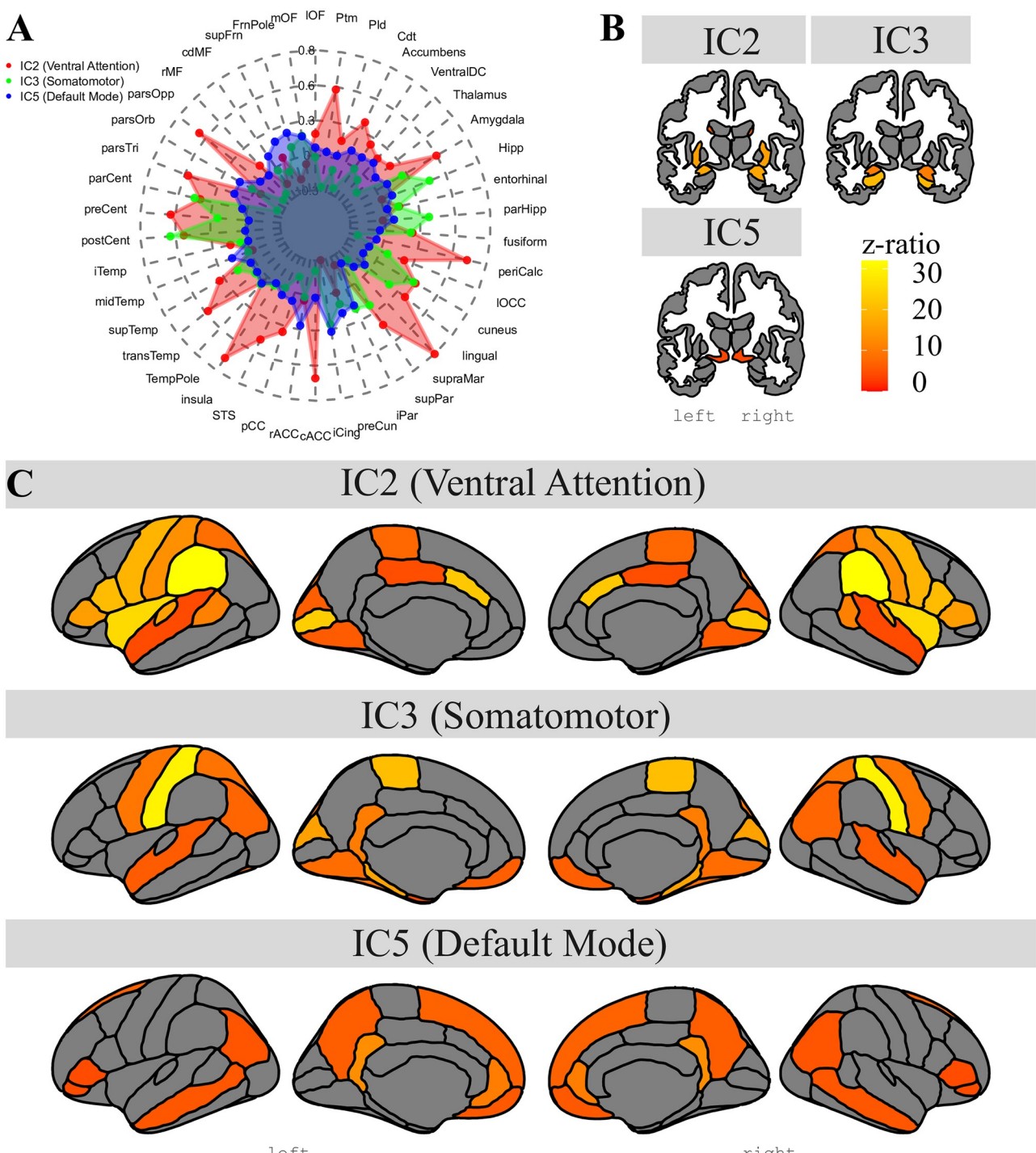

**Fig 5. Results from group-level analyses for all ROIs.** Representation of estimated marginal mean group-level FC values between IC hotspots inside the amygdala and each cortical and subcortical region using a spiderplot (A), as well as the z-ratios representing group-level FC between IC hotspots inside the amygdala and subcortical (B) and cortical (C) regions. Note how different hotspots of activity detected inside the amygdala are functionally connected with clearly different whole-brain networks.

**Table 4. Cortical and subcortical areas showing reliable co-activity with IC2 (correlated with ventral attention network) relative to the mean co-activity value of all other areas.** p-values corrected for multiple comparisons using Bonferroni correction.

| Region | Z-ratio | p-value |
|---|---|---|
| supramarginal | 31.99 | 7.403E-223 |
| insula | 25.55 | 2.136E-142 |
| pericalcarine | 23.49 | 2.173E-120 |
| caudal anterior cingulate | 22.33 | 7.522E-109 |
| pars opercularis | 20.44 | 3.361E-91 |
| precentral | 18.99 | 9.397E-79 |
| amygdala | 16.71 | 5.183E-61 |
| putamen | 15.74 | 3.271E-54 |
| transverse temporal | 15.69 | 8.081E-54 |
| pars triangularis | 15.37 | 1.127E-51 |
| postcentral | 13.33 | 6.645E-39 |
| bankssts | 11.04 | 1.065E-26 |
| superior parietal | 8.36 | 2.695E-15 |
| paracentral | 7.36 | 8.015E-12 |
| cuneus | 6.39 | 7.479E-09 |
| lingual | 5.93 | 1.329E-07 |
| caudate | 5.87 | 1.906E-07 |
| posterior cingulate | 4.47 | 3.387E-04 |
| superior temporal | 4.12 | 1.655E-03 |

nine nuclei revealed the relative contribution of each amygdala nucleus. Specifically, as can be seen in Table 7, for the FC map correspond to IC2 (correlated with the ventral attention network) summed z-ratios were in descending order ranked Co, AAA, CAT and Ce. Similarly, Table 7 showed that for FC map associated with IC3 (correlated with the somatomotor

**Table 5. Cortical and subcortical areas showing reliable co-activity with IC3 (correlated with Somatomotor network) relative to the mean co-activity value of all other areas.** p-values corrected for multiple comparisons using Bonferroni correction.

| Region | Z-ratio | p-value |
|---|---|---|
| postcentral | 29.89 | 1.268E-194 |
| paracentral | 20.65 | 4.622E-93 |
| hippocampus | 20.10 | 2.936E-88 |
| parahippocampal | 16.40 | 8.977E-59 |
| cuneus | 16.14 | 5.890E-57 |
| amygdala | 10.43 | 8.132E-24 |
| isthmus cingulate | 10.39 | 1.206E-23 |
| precentral | 9.57 | 4.534E-20 |
| superior parietal | 8.94 | 1.707E-17 |
| fusiform | 8.50 | 8.452E-16 |
| lingual | 8.08 | 2.960E-14 |
| inferior parietal | 7.12 | 4.902E-11 |
| medial orbitofrontal | 6.39 | 7.221E-09 |
| superior temporal | 5.94 | 1.241E-07 |
| entorhinal | 3.84 | 5.370E-03 |

**Table 6. Cortical and subcortical areas showing reliable co-activity with IC5 (correlated with default mode network) relative to the mean co-activity value of all other areas.** p-values corrected for multiple comparisons using Bonferroni correction.

| Region | Z-ratio | p-value |
|---|---|---|
| isthmus cingulate | 13.06 | 2.408E-37 |
| rostral anterior cingulate | 10.36 | 1.687E-23 |
| frontal pole | 8.54 | 5.752E-16 |
| superior frontal | 6.90 | 2.284E-10 |
| precuneus | 6.27 | 1.550E-08 |
| medial orbitofrontal | 5.66 | 6.660E-07 |
| middle temporal | 5.33 | 4.392E-06 |
| inferior parietal | 5.08 | 1.634E-05 |
| pars orbitalis | 4.33 | 6.433E-04 |
| pars triangularis | 4.28 | 8.365E-04 |
| ventral DC | 3.66 | 1.091E-02 |

network), summed z-ratios were ranked CAT, PL, AAA, AB and Ba. Finally, Table 7 showed that for the FC map linked to IC11 (correlated with the default mode network) summed z-ratios were ranked Co, AB, CAT, Me and Ce (see also Fig 6A and 6B for a graphical presentation of these results).

## Validation results

Computation of the whole-brain FC maps for the two additional resting-state volumes in the validation dataset using the seed clusters detected inside the amygdala from the test set revealed a set of results in the validation set that was highly similar to those obtained in the test set (see S4 Fig). Direct comparisons of the whole-brain FC maps from the test and validation datasets corresponding to ICs 2, 3 and 5 yielded Pearson's correlations of 0.95, 0.87 and 0.81, respectively. Finally, correlation of these three whole-brain FC maps with the reference Yeo networks yielded similar degrees of overlap with the same networks as detected in the test set

**Table 7. Relative contributions of each amygdaloid subnucleus within the different resting-state networks.** Rankings based on the sum of pairwise z-ratio differences for all subnuclei within a given IC.

| Subnucleus | IC | Network | summed z-ratio | rank |
|---|---|---|---|---|
| Co | 2 | Ventral Attention | 181.45 | 1 |
| AAA | 2 | | 156.99 | 2 |
| CAT | 2 | | 63.84 | 3 |
| Ce | 2 | | 5.49 | 4 |
| CAT | 3 | Somatomotor | 262.72 | 1 |
| PL | 3 | | 154.54 | 2 |
| AAA | 3 | | 111.72 | 3 |
| AB | 3 | | 16.89 | 4 |
| Ba | 3 | | 8.11 | 5 |
| Co | 5 | Default Mode | 202.22 | 1 |
| AB | 5 | | 67.67 | 2 |
| CAT | 5 | | 34.98 | 3 |
| Me | 5 | | 22.10 | 4 |
| Ce | 5 | | 7.67 | 5 |

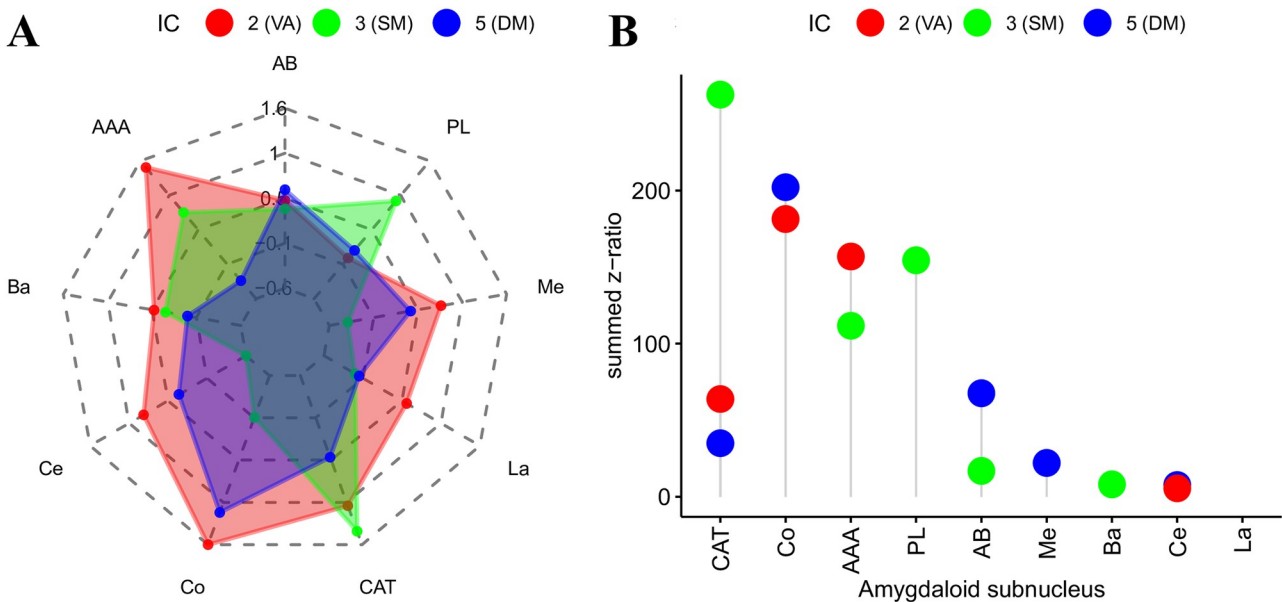

**Fig 6. Results from group-level analyses for amygdala nuclei.** Estimated marginal mean co-activity values for each amygdaloid nucleus across the three IC hotspots detected in the amygdala (A), as well as the relative contributions of each amygdaloid subnucleus within each IC hotspot in terms of summed group-level pairwise comparisons (B). Note how the different IC hotspots that are associated with different resting-state networks rely on different configurations of subnucleus co-activity.

(IC2 with VA, $r = 0.47$; IC3 with SM, $r = 0.40$; IC5 with DMN, $r = 0.65$, compare with Table 3). We can therefore conclude that the clusters of activity detected inside the amygdala are robust and generalize to different datasets.

## Impact of self-reported fear on amygdaloid nucleus co-activity

Statistical analysis revealed a main effect of Gender ($F(1,169)=4.70$, $p < 0.04$), indicating larger co-activity values in the amygdala for women than for men. There was no effect of Age, which therefore was removed from the model. There was an effect of Hemisphere ($F(1,9063)=296.76$, $p < 0.0001$), indicating higher co-activity values for the right hemisphere. Furthermore, there was an effect of brain region ($F(8,9063)=53.22$, $p < 0.0001$), suggesting differences in co-activity values between the nine nuclei. There was also an effect of FC map ($F(2,9063)=132.13$, $p < 0.0001$), suggesting differences in co-activity between the three FC maps. As before, there was an interaction between brain region and FC map ($F(16,9063)=46.85$, $p < 0.0001$), suggesting the co-activity values for the various nuclei differed between FC maps. Finally, and important for our present purposes, there was an interaction between Group and brain region ($F(8,9063)=2.07$, $p < 0.04$), suggesting that the effect of group was dependent on the brain region. The triple interaction between Group, Brain Region and FC map was not significant. Further exploration of this two way interaction revealed that co-activity values in the high fear group were higher than those in the low fear group for the Co nucleus (z-ratio = 3.322, $p < 0.001$). Other comparisons were not significant. A visual presentation of these results is shown in Fig 7B.

## Discussion

The current study examined the contributions of nine amygdaloid nuclei to different resting-state networks. The results revealed that the amygdala relied on three distinct resting-state

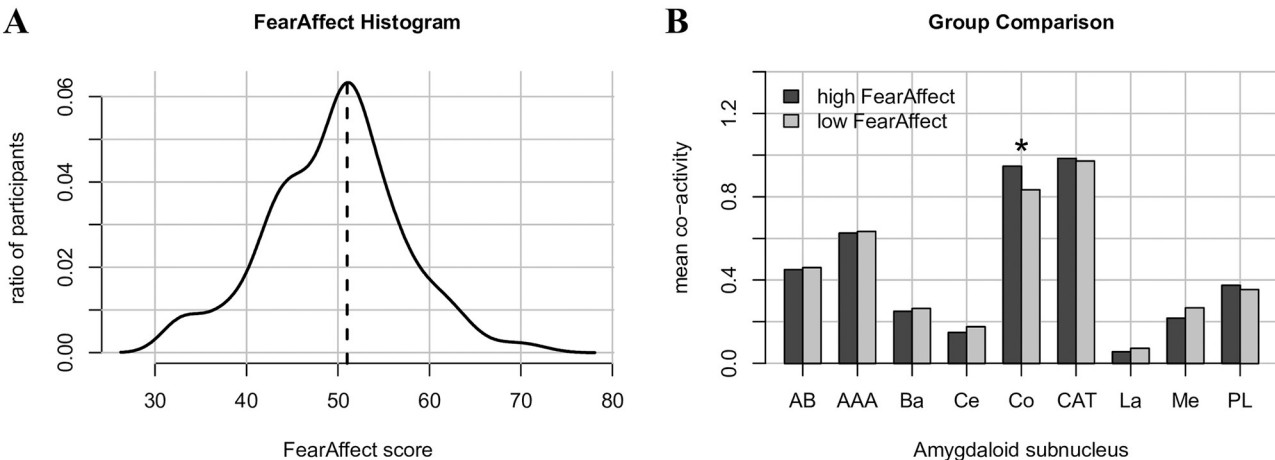

**Fig 7. Results from fear affect analyses.** Distribution of Fear Affect scores (A) with low and high FearAffect groups defined by the median value (indicated by the vertical line), and marginal means of co-activity scores between the low and high fear group for each amygdaloid nucleus. * indicates significant difference (p < 0.001).

networks and that each network relied on a specific configuration of amygdaloid nuclei. Specifically, the somatomotor network (*r* = 0.45 with reference network) relied on the CorticoAmygdaloid Transition area (CAT), Paralaminar (PL), Anterior Amygdaloid Area (AAA), Accessory Basal (AB) and Basal (Ba) nuclei. In addition, the ventral attention network (*r* = 0.48 with reference network) relied on the Cortical (Co), AAA, CAT and Central (Ce) nuclei. Furthermore, the default mode network (*r* = 0.59 with reference network) was co-activated with the Co, AB, CAT, Medial (Me) and Ce nuclei. These results were validated in a separate dataset. Finally, we also found that participants' ratings of self-reported fear were associated with co-activity in the Co nucleus.

These results challenge the classical organization of the amygdala in terms of three large-scale divisions as mentioned in the Introduction (LB, CM, and SF; [23]). First, for the somatomotor network (co-activated with CAT, PL, AAA, AB and Ba nuclei), the majority of the activated nuclei belong to the LB division (3/5), while CAT and AAA belong to the SF division (see Table 1 for reference). These results are generally consistent with previous FC studies that have linked the LB and SF sections of the amygdala with primary sensory regions like postcentral gyrus, STG and olfactory cortex [6, 25, 26, 51, 52]. Second, for the ventral attention network (co-activated with Co, AAA, CAT and Ce nuclei), the majority of these nuclei (3/4) can be classified as belonging to the SF division and the Ce nucleus to the CM division. As mentioned in the Introduction, previous studies have not been able to clearly associate particular large-scale amygdala sections with the ventral attention network [6]. Finally, for the default mode network (co-activated with Co, AB, CAT, Me and Ce nuclei), where nuclei belong to the SF (2/5), CM (2/5) and LB (1/5) divisions. Again, this result is difficult to relate to previous human FC studies because they have not found consistent associations between a specific large-scale subdivision of the amygdala and the default mode network (cf., [6, 26, 51]). The current results show that there are three activity clusters in the amygdala that were strongly and uniquely correlated with different whole-brain resting-state networks. Importantly, these three activity clusters cut across the traditional LB, CM and SF subdivisions [23].

The observation that the amygdala is related to somatomotor, ventral attention, and default mode networks corroborates previous conclusions about the functional role of the amygdala. First, the observation that the amygdala is related to the somatomotor network is consistent with the overall conception of the amygdala as a structure involved in the on-line evaluation of

incoming sensory information (external or internal) in affective terms [53, 54]. Our findings therefore suggest that the CAT, PL, AAA, AB and Ba amygdala nuclei are especially involved in processing of sensorimotor information. In addition, the involvement of the amygdala with the ventral attention network is consistent with recent findings in both animals and humans that amygdala activity is modulated by the spatial location of an emotionally charged stimulus [55–58]. The results reported in the current study implicate the Co, AAA, CAT and Ce amygdala nuclei in the ventral attention network and therefore highlight these nuclei for further examination of the relationship between attention and the amygdala. Finally, altered amygdala connectivity with nodes of the default mode network has been frequently reported in the context of psychiatric disorders (e.g., [59, 60]). For example, it is now relatively well understood that generalized anxiety disorders affect connectivity between the amygdala and the ventromedial prefrontal cortex, a key node in the default mode network [16, 61, 62]. Our finding that the Co, AB, CAT, Me and Ce amygdaloid nuclei were co-activated with the default mode network calls attention to the potential role of these nuclei in various psychiatric disorders such as anxiety and depression.

A more complex issue concerns a direct comparison of our results to those obtained in the animal literature. On the one hand, as mentioned in the Introduction, studies of the anatomical connectivity of the amygdala in non-human animals suggest that different brain regions may project to multiple amygdala nuclei [17]. Thus, for example, areas like the prefrontal cortex and the hippocampus are known to project to multiple amygdala nuclei [18–20, 22]. These observations fit well with the finding that a given resting-state network relied on a complex configuration of amygdaloid nuclei that is not confined to the classic division. On the other hand, as mentioned earlier, studies of the function of the different amygdala nuclei in rodent studies have associated separate functions with separate nuclei ([7], but see [63]). One possibility is that these differences stem from the more constrained experimental tasks used in rodent studies compared to the complex mental states that presumably underlie the various resting-state networks in humans. In this context, an interesting observation in the current study was the absence of La nucleus co-activity. However, given the association between the La nucleus and direct sensory input, this absence may also reflect the resting-state design which does not emphasize sensory input. Future studies with humans that examine the amygdala nuclei with task-based fMRI could shed further light on this issue.

A pertinent observation in the current study is the relevance of the Co nucleus. This nucleus was strongly co-activated with two of the three observed resting-state networks (Default Mode Network and Ventral Attention network; see Fig 6), and the co-activation of this nucleus was affected by the self-reported fear of participants (see Fig 7). Previous studies in non-human animals have suggested that the cortical nucleus is primarily involved in olfactory processing, and that more posterior sections of this nucleus are linked to memory processes due to connectivity with structures in the medial temporal lobe [64–66]. In humans, a number of recent studies have implicated structural changes in the Co nucleus with mood and anxiety related disorders [1, 2, 13, 14]. For example, [14] found that risk for mental illness significantly modulated the volumes of left medial, left central and bilateral cortical nuclei in the amygdala. Similarly, [13] found that the major depressive disorder was associated with increased white matter connectivity from the lateral, basal, central and cortical nuclei to the rest of the brain. Thus, our observations that the Co nucleus co-activates with various whole-brain networks as well as its involvement in self-reported fear is in line with these previous findings. Taken together, the results from our study and those previously observed on the involvement of the Co nucleus underscore the need to re-evaluate its function and to consider this nucleus as a target for future clinical research.

## Conclusion

To conclude, in the current study we examined the contributions of nine amygdala nuclei to existing resting-state networks using a high spatial resolution rsfMRI dataset. Our findings show that the amygdala co-activates with three resting-state networks: The somatomotor, ventral attention and default mode network [6]. In addition, we show that these three networks rely on distinct configurations of co-activity across the nine amygdaloid nuclei that cut across the traditional boundaries between the LB, CM and SF divisions [23]. These results are in line with animal studies that underscore the complex nature of amygdala connectivity where multiple nuclei connect with many other brain regions [17, 63]. Finally, the results highlight a special role for the Co nucleus in that this nucleus was most strongly co-activated with the ventral attention and default mode networks, and that its co-activation was associated with the self-reported fear of the study participants. Taken together these results revealed the FC of the amygdala nuclei in a healthy human population and may serve as a baseline against which to evaluate pathological conditions examined in future studies.

## Supporting information

**S1 Fig. Overview of correlations between whole-brain FC maps and the 7 Yeo networks for each IC across all 15 dimensions tested.** Note that three Yeo networks (2,4,7) appear frequently across all dimensions and that the lowest dimension at which these three networks are detected with good strength and separability is dimension 7.
(TIF)

**S2 Fig. Correlations between independent components derived from srICA at dimensions 20 and 30 and the 7 different resting-state networks.** Note how even at increased dimensions, networks 2 (somatomotor), 4 (ventral attention) and 7 (default mode) seem most frequently detected, albeit less reliably.
(TIF)

**S3 Fig. Overview of the spatial location of detected ICs inside the amygdala (left column, A), as well as their corresponding whole-brain FC maps (right column, B) for all ICs with dimension 7.** Note ICs 2 (Ventral Attention), 3 (Somatomotor), and 5 (default mode) were detected by the algorithm as having a strong and unique relationship with the Yeo networks (see main manuscript text for details).
(TIF)

**S4 Fig. Overview of results in the validation dataset.** Note the high similarity between the obtained functional connectivity for ICs 2, 3 and 5 in the validation and test datasets (cf., S3 Fig).
(TIF)

**S1 File.**
(PDF)

## Acknowledgments

## Author Contributions

**Conceptualization:** Uriel K. A. Elvira, Sara Seoane, Joost Janssen, Niels Janssen.

**Data curation:** Niels Janssen.

**Formal analysis:** Sara Seoane, Niels Janssen.

**Methodology:** Uriel K. A. Elvira, Sara Seoane, Niels Janssen.

**Project administration:** Niels Janssen.

**Validation:** Niels Janssen.

**Visualization:** Uriel K. A. Elvira, Sara Seoane, Niels Janssen.

**Writing – original draft:** Uriel K. A. Elvira, Sara Seoane, Joost Janssen, Niels Janssen.

**Writing – review & editing:** Uriel K. A. Elvira, Joost Janssen, Niels Janssen.

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
