## [Decision Letter · Decision Letter 0]

11 Oct 2022

PONE-D-22-24147Contributions of human amygdala nuclei to resting-state networksPLOS ONE

Dear Dr. Janssen,

Thank you for submitting your manuscript to PLOS ONE. After careful consideration, we feel that it has merit but does not fully meet PLOS ONE’s publication criteria as it currently stands. Therefore, we invite you to submit a revised version of the manuscript that addresses the points raised during the review process.

We look forward to receiving your revised manuscript.

Kind regards,

Irene Cristofori

Academic Editor

PLOS ONE

**Journal Requirements: **

2. In the Methods section and the online submission form, please provide additional information about the patient records used in your retrospective study. Specifically, please ensure that you have discussed whether all data were fully anonymized before you accessed them and/or whether the IRB or ethics committee waived the requirement for informed consent. If patients provided informed written consent to have data from their medical records used in research, please include this information.

"Data were provided by the Human Connectome Project, WU-Minn Consortium (Principal Investigators: David Van Essen and Kamil Ugurbil; 1U54MH091657) funded by the 16 NIH Institutes and Centers that support the NIH Blueprint for Neuroscience Research; and by the McDonnell Center for Systems Neuroscience at Washington University. This work was supported by grants PSI2017-84933-P and PSI2017-91955-EXP to NJ. SLS was supported by a pre-doctoral fellowship from the Canarian Agency for Research, Innovation, Society and Information (ACIISI) and by a graduate grant from the Santander Bank Foundation at the University of La Laguna. The funders had no role in study design, data collection and analysis, decision to publish, or preparation of the manuscript."

"No"

Reviewer comments:

**Comments to the Author**

1. Is the manuscript technically sound, and do the data support the conclusions?

Reviewer #1: Yes

Reviewer #2: Yes

2. Has the statistical analysis been performed appropriately and rigorously? 

Reviewer #1: Yes

Reviewer #2: Yes

3. Have the authors made all data underlying the findings in their manuscript fully available?

Reviewer #1: Yes

Reviewer #2: Yes

4. Is the manuscript presented in an intelligible fashion and written in standard English?

Reviewer #1: Yes

Reviewer #2: Yes

5. Review Comments to the Author

Reviewer #1: The work by Arguinzones et al is an interesting contribution to the functional connectivity of the different amygdaloid nuclei in resting conditions. Surprinsingly, to my view, the contributions of the different nuclei to the functional networks identified do not correspond to the classical amygdaloid subdivisions. These subdivision are defined mainly based on work in rodents and (to a lesser extent) in non-human primates.

I have only minor comments:

- In the abstract, the abbreviations of the amygdaloid subnuclei are not that common to be identified by the readers. I suggest either spelling them out at first use or avoid using abbreviations in the abstract.

- In the Discussion, it may be worth to include a few lines about possible limitations of the study or limitations with the comparisond with the animal data, which may contribute to explain such important differences with previously defined functional subdivisions. For example, I am not really sure to what extent the medial amygdala or the anterior amygdaloid area in humans are the same structures as those identified with the same names in rodents. Something similar with the cortical nucleus, which in rodents is divided in two olfactory nuclei (anterior cortical and posterolateral cortical) and one vomeronasal (pheromonal) nucleus (posteromedial cortical). To what extent is the functional role of the cortical nucleus in humans comparable to these cortical nuclei in rodents is not an easy question.

A different kind of caveat is the small size of some of the structures, such as the medial amygdala (as represented in Fig 1). How is this affecting to the data? I guess that the results in this nucleus may not be as robust as those in the larger nuclei.

Finally, the fact that the lateral nucleus, which is the largest nucleus in the human amygdala and one of the best characterized in the literature in non-human animals, is not involved in any of the functional networks may also deserve a word in the Discussion.

Page 14, line 5 from the bottom: "contribution each amygdala nucleus" should probably be "contribution OF each amygdala nucleus"

Reviewer #2: The study used a data-driven approach to investigate the functional connectivity (FC) patterns of the amygdaloid nuclei with existing whole-brain resting state networks. The methods are sound and the results interesting. However, in my opinion, the manuscript needs substantial work in the introduction.

A description of the functional role of the amygdala (and the amygdaloid nuclei) in the introduction is completely lacking. For example, when listing the main objectives of their work, the authors mention “In addition, we tested whether the self-reported fear of the participants was associated with co-activity in the amygdaloid nuclei”. This comes unexpected, without any description of why this is relevant. What is the proposed functional role of the amygdala and its sub-regions/nuclei? What does self-reported fear mean? Why self-reported fear should be associated with FC of the amygdala? What are the specific hypotheses/predictions concerning the different amygdaloidal nuclei? Please expand the introduction according to these relevant questions. This will allow you to make the introduction more coherent with the discussion, where possible implications of the current results in the clinical field are discussed (but should be maybe more specific).

It is not clear if the spatial resolution of fMRI data used in previous studies was lower with respect to HCP data or simply no previous study looked at the single nuclei.

ICA allowed to identify 3 seeds within the amygdala region. Do they overlap, at least partially, with the large-scale sections previously considered (LB, CM and SF)?

rsfMRI, QC, CSF and WM. Please spell out all acronyms the first time you use them.

Page 7. Last lines. “These subdivisions do not take into account the concept of the "extended amygdala". Please explain.

Please provide additional details for the segmentation method: “by combining prior information about the average structural location of the amygdaloid nuclei and its surrounding tissue with specific structural information from a given participant’s brain”. What specific structural information? For example?

Page 11, first line: “Finally, we explored whether self-reported measures of fear affected co-activity of the amygdala nuclei”. I would rather talk about “exploring a possible relationship between fear and FC”, without further hypothesis about the direction of the effect. I don’t think we can assume that behavior affects FC.

6. PLOS authors have the option to publish the peer review history of their article (what does this mean?). If published, this will include your full peer review and any attached files.

Reviewer #1: No

Reviewer #2: No

---

## [Author Response · Author response to Decision Letter 0]

4 Nov 2022

Reviewer #1: The work by Arguinzones et al is an interesting contribution to the functional connectivity of the different amygdaloid nuclei in resting conditions. Surprinsingly, to my view, the contributions of the different nuclei to the functional networks identified do not correspond to the classical amygdaloid subdivisions. These subdivision are defined mainly based on work in rodents and (to a lesser extent) in non-human primates.

I have only minor comments:

- In the abstract, the abbreviations of the amygdaloid subnuclei are not that common to be identified by the readers. I suggest either spelling them out at first use or avoid using abbreviations in the abstract.

We thank the reviewer for this comment. We have rewritten the abstract and improved the abbreviations for nuclei throughout the manuscript.

- In the Discussion, it may be worth to include a few lines about possible limitations of the study or limitations with the comparisons with the animal data, which may contribute to explain such important differences with previously defined functional subdivisions. For example, I am not really sure to what extent the medial amygdala or the anterior amygdaloid area in humans are the same structures as those identified with the same names in rodents. Something similar with the cortical nucleus, which in rodents is divided in two olfactory nuclei (anterior cortical and posterolateral cortical) and one vomeronasal (pheromonal) nucleus (posteromedial cortical). To what extent is the functional role of the cortical nucleus in humans comparable to these cortical nuclei in rodents is not an easy question. A different kind of caveat is the small size of some of the structures, such as the medial amygdala (as represented in Fig 1). How is this affecting to the data? I guess that the results in this nucleus may not be as robust as those in the larger nuclei.

We thank the reviewer for these crucial insights. We agree with the reviewer that overall comparisons between humans and animal studies are not straightforward and we believe that future studies that examine the amygdala nuclei with task fMRI (instead of resting-state fMRI) may be beneficial in resolving some of these questions. We discuss these issues in a new paragraph in the general discussion where we go into detail about the comparison between our results and those obtained in non-human animal studies. 

See page 16, line 444

Finally, the fact that the lateral nucleus, which is the largest nucleus in the human amygdala and one of the best characterized in the literature in non-human animals, is not involved in any of the functional networks may also deserve a word in the Discussion.

We thank the reviewer for pointing this out. In the added paragraph we also speculate as to why we did not find activity in the LA nucleus. In our opinion this may be related to the nature of the resting-state study that we conducted which by definition only relies on minimal amounts on sensory input. Under such conditions no LA activity may be observed. As we also mentioned before, fMRI studies using tasks (which require strong sensory input) may be able to shed further light on this issue. We again thank the reviewer for bringing up this inspiring point. 

See page 16, line 444

Page 14, line 5 from the bottom: "contribution each amygdala nucleus" should probably be "contribution OF each amygdala nucleus"

Fixed.

Reviewer #2: The study used a data-driven approach to investigate the functional connectivity (FC) patterns of the amygdaloid nuclei with existing whole-brain resting state networks. The methods are sound and the results interesting. However, in my opinion, the manuscript needs substantial work in the introduction.

A description of the functional role of the amygdala (and the amygdaloid nuclei) in the introduction is completely lacking. For example, when listing the main objectives of their work, the authors mention “In addition, we tested whether the self-reported fear of the participants was associated with co-activity in the amygdaloid nuclei”. This comes unexpected, without any description of why this is relevant. What is the proposed functional role of the amygdala and its sub-regions/nuclei? What does self-reported fear mean? Why self-reported fear should be associated with FC of the amygdala? What are the specific hypotheses/predictions concerning the different amygdaloidal nuclei? Please expand the introduction according to these relevant questions. This will allow you to make the introduction more coherent with the discussion, where possible implications of the current results in the clinical field are discussed (but should be maybe more specific).

We thank the reviewer for bringing this up. We have now largely rewritten the Introduction and parts of the Discussion to accommodate the reviewer’s concerns.

It is not clear if the spatial resolution of fMRI data used in previous studies was lower with respect to HCP data or simply no previous study looked at the single nuclei.

Indeed the spatial resolution in previous studies was lower (2 to 4 mm) compared to that of the HCP (1.6 mm). It is likely that previous studies have not looked at amygdala nuclei because of this insufficient spatial resolution. 

We hope to have clarified this issue in the current manuscript. 

ICA allowed to identify 3 seeds within the amygdala region. Do they overlap, at least partially, with the large-scale sections previously considered (LB, CM and SF)?

We thank the reviewer for highlighting this important point. We have largely rewritten the first two paragraphs of the Discussion section to better highlight this issue in the current version of the manuscript. In the current manuscript we highlight that the relationship between the specific clusters we have found (and the nuclei that are involved in each cluster) and the classical division of the amygdala into LB, CM and SF sections. We hope to have improved our manuscript with respect to this issue.

rsfMRI, QC, CSF and WM. Please spell out all acronyms the first time you use them.

Fixed.

Page 7. Last lines. “These subdivisions do not take into account the concept of the "extended amygdala". Please explain.

The extended amygdala refers to the bed nucleus of the stria terminalis (BNST) in the basal forebrain. There is debate in the literature about whether to consider this structure as part of the amygdala (e.g., Lebow & Chen, 2016). The current parcellation of the amygdala into its constituent nuclei (Saygin et al., 2017) does not include this structure.

We have added some words to better explain this issue. We thank the reviewer for pointing this out. 

Methods page 5, line 177

Please provide additional details for the segmentation method: “by combining prior information about the average structural location of the amygdaloid nuclei and its surrounding tissue with specific structural information from a given participant’s brain”. What specific structural information? For example?

Specific structural information that is used is the relative position of the surrounding structures (coming from the aparc+aseg file) as well as the specific intensity values present in the T1w file. 

We have updated the manuscript to better explain this issue. 

Methods, page 5, line 180

Page 11, first line: “Finally, we explored whether self-reported measures of fear affected co-activity of the amygdala nuclei”. I would rather talk about “exploring a possible relationship between fear and FC”, without further hypothesis about the direction of the effect. I don’t think we can assume that behavior affects FC.

We completely agree with the reviewer about this issue and we apologize for the slip up. We did not wish to imply any causality here. We have rewritten these sections. 

We thank the reviewer for the insightful comments!

---

## [Decision Letter · Decision Letter 1]

25 Nov 2022

Contributions of human amygdala nuclei to resting-state networks

PONE-D-22-24147R1

Dear Dr. Janssen,

We’re pleased to inform you that your manuscript has been judged scientifically suitable for publication and will be formally accepted for publication once it meets all outstanding technical requirements.

Kind regards,

Irene Cristofori

Academic Editor

PLOS ONE

Additional Editor Comments (optional):

Reviewers' comments:

Reviewer's Responses to Questions

**Comments to the Author**

1. If the authors have adequately addressed your comments raised in a previous round of review and you feel that this manuscript is now acceptable for publication, you may indicate that here to bypass the “Comments to the Author” section, enter your conflict of interest statement in the “Confidential to Editor” section, and submit your "Accept" recommendation.

Reviewer #1: All comments have been addressed

Reviewer #2: All comments have been addressed

2. Is the manuscript technically sound, and do the data support the conclusions?

Reviewer #1: Yes

Reviewer #2: Yes

3. Has the statistical analysis been performed appropriately and rigorously? 

Reviewer #1: Yes

Reviewer #2: Yes

4. Have the authors made all data underlying the findings in their manuscript fully available?

Reviewer #1: Yes

Reviewer #2: Yes

5. Is the manuscript presented in an intelligible fashion and written in standard English?

Reviewer #1: Yes

Reviewer #2: Yes

6. Review Comments to the Author

Reviewer #1: The authors have adequately addressed the reviewers' comments. I have no further suggestions.

Reviewer #2: The authors have addressed all the comments I made during the first revision run. I recommend the manuscript for publication.

7. PLOS authors have the option to publish the peer review history of their article (what does this mean?). If published, this will include your full peer review and any attached files.

Reviewer #1: **Yes: **Enrique Lanuza

Reviewer #2: No

---

## [Editor Report · Acceptance letter]

14 Dec 2022

PONE-D-22-24147R1 

Contributions of human amygdala nuclei to resting-state networks 

Dear Dr. Janssen:

I'm pleased to inform you that your manuscript has been deemed suitable for publication in PLOS ONE. Congratulations! Your manuscript is now with our production department. 

Kind regards, 

on behalf of

Dr. Irene Cristofori 

Academic Editor

PLOS ONE